# Biomarkers and Hemodynamic Parameters in the Diagnosis and Treatment of Children with Postural Tachycardia Syndrome and Vasovagal Syncope

**DOI:** 10.3390/ijerph19126974

**Published:** 2022-06-07

**Authors:** Wenjie Cheng, Jiaqi Wang, Jing Lin

**Affiliations:** 1School of Public Health, Xi’an Jiaotong University, Xi’an 710061, China; chengwj20@stu.xjtu.edu.cn (W.C.); a568317505@163.com (J.W.); 2Department of Maternal and Child Health, School of Public Health, Xi’an Jiaotong University, Xi’an 710061, China

**Keywords:** orthostatic intolerance, vasovagal syncope, postural tachycardia syndrome, differential diagnosis, individualized treatment

## Abstract

In children, vasovagal syncope and postural tachycardia syndrome constitute the major types of orthostatic intolerance. The clinical characteristics of postural tachycardia syndrome and vasovagal syncope are similar but their treatments differ. Therefore, their differential diagnosis is important to guide the correct treatment. Therapeutic methods vary in patients with the same diagnosis because of different pathomechanisms. Hence, in patients with vasovagal syncope or postural tachycardia syndrome, routine treatments have an unsatisfactory efficacy. However, biomarkers could increase the therapeutic efficacy significantly, allowing for an accurate and detailed assessment of patients and leading to improved therapeutic effects. In the present review, we aimed to summarize the current state of research into biomarkers for distinguishing the diagnosis of pediatric vasovagal syncope from that of postural tachycardia syndrome. We also discuss the biomarkers that predict treatment outcomes during personalized therapy for each subtype.

## 1. Introduction

The inability to tolerate the upright posture is referred to as orthostatic intolerance (OI) and comprises a series of clinical symptoms including dizziness, headache, and temporary loss of consciousness. OI can be relieved after recumbency [1], occurs frequently, and affects both the quality of life and psychosocial health [2]. OI pathogenesis is mainly associated with autonomic dysfunction, central hypovolemia, an abnormal Bezold–Jarisch reflex, and an abnormal endothelium-dependent diastolic function [3,4,5,6]. In children and adolescents, VVS (vasovagal syncope) and POTS (postural tachycardia syndrome) are responsible for 70–80% of OI [7,8]. The clinical signs of POTS and VVS are similar but their pathogeneses are different, thus necessitating different treatments; however, care should be taken to distinguish the subtypes. The current accepted criteria to diagnose POTS and VVS in children comprise a combination of clinical data and clinical symptoms observed during a head-up tilt test (HUTT). However, a HUTT may cause episodes of syncope or asystole, usually leading to discomfort among children and adding to their psychological loads, and its widespread clinical application is thus restricted [9]. Therefore, novel, acceptable, safe, and simple criteria are required to diagnose POTS and VVS in children.

The mechanisms for VVS and POTS remain unclear. Their pathogeneses are believed to be related to the impaired regulation of peripheral vascular resistance, autonomic nervous system imbalance, hyper-adrenergic responses, and absolute hypovolemia. Consequently, children suffering from VVS or POTS might be treated using water, β-blockers, salt, or midodrine. However, the efficacy of the drugs varies.

Recently, biomarkers or certain hemodynamic parameters that can predict the treatment effects of individualized treatment for POTS or VVS have been used. Hence, this article summarizes the physiological indicators used to differentially diagnose POTS and VVS and their individualized treatment in children and discusses further research directions in this field.

## 2. Differential Diagnosis of POTS and VVS

Despite their similar clinical manifestations, different methods and strategies are used to treat VVS and POTS. A HUTT can be used to diagnose both but it can be very uncomfortable and in rare cases, it can cause arrhythmias or cardiac arrest. Currently, non-invasive differential diagnosis is an important clinical issue in this field. Therefore, finding a sensitive and reliable method for differential diagnosis between the two diseases has become an urgent clinical need. An investigation of the physiological indicators that differ between VVS and POTS could effectively improve diagnosis, which is of great significance for clinical diagnoses and precise treatments. In this review, firstly, we summarize the physiological indicators used to deferentially diagnose POTS and VVS (Table 1).

### 2.1. The Plasma Hydrogen Sulfide (H_2_S) Level

The toxic gas hydrogen sulfide (H_2_S) was recognized recently as an endogenous gasotransmitter [10]. H_2_S contributes to endothelium-dependent vasorelaxation and exerts regulatory effects on the pathogenesis of various diseases [11]. According to Zhang et al., the plasma concentration of H_2_S can help distinguish between POTS and VVS in children. The results of that study indicated that children with POTS and VVS had higher H_2_S plasma levels than healthy children. A plasma level of H_2_S at 98 μmol/L taken as the cutoff value produced both high sensitivity (90%) and specificity (80%) rates for correctly discriminating between patients with VVS and patients with POTS [12].

### 2.2. The Serum Iron Level

Generally, VVS is rare or sporadic, whereas POTS represents a chronic daily form of OI. However, VVS patients sometimes experience chronic OI symptoms and POTS patients might experience a temporary or sudden loss of consciousness. Hence, based on symptoms alone, the differential diagnosis of VVS from POTS is often difficult. Patients with POTS or VVS have a high prevalence of chronic fatigue. Iron deficiency was proven to be associated with chronic fatigue in patients with OI [13,14]. Interestingly, the symptoms of OI could be relieved using iron supplementation or the administration of recombinant erythropoietin [15,16]. Thus, the mechanisms were probably linked to the oxygen-carrying capacity of hemoglobin and the serum iron levels might be different between POTS and VVS. According to a study by Li et al., the serum iron level was higher among POTS children than among children with VVS (with significant differences in their median values), which could be used as a preliminary method to differentiate POTS from VVS in a clinic. When the value of serum iron was 11.8 μmol/L, VVS could be distinguished from POTS with 92.5% sensitivity and 64.7% specificity [17].

### 2.3. Immediate Heart Rate Alteration Index AI and 30/15

The instantaneous HR (heart rate) variation from the supine position to standing can be represented by the 30/15 ratio and the AI (acceleration index). The AI is calculated using the following equation: AI = ((A − B)/A) × 100, where A is the average duration of the R-R interval during the 15 s prior to the position change, and B denotes the initial shortest R-R interval following the position change [18]. The length of the R-R interval for the 30th beat as a percentage of that for the 15th beat in the upright position is 30/15 [19]. Both ratios are associated with cardiovascular and autonomic nervous functions. Tao et al. investigated using the value of AI and 30/15 to differentially diagnose VVS and POTS. Compared with children with VVS, the AI was prominently higher in POTS children and the 30/15 was lower. Thus, both ratios might be useful to differentially diagnose VVS and POTS. For AI, using a cut-off value of 28.180 resulted in 79.2% sensitivity and 73.1% specificity. The adoption of a 1.025 threshold for 30/15 resulted in 87.5% sensitivity and 61.5% specificity. Using both ratios jointly, the sensitivity was elevated to 95.8% and the specificity to 80.8% for diagnosis [20].

### 2.4. Frequency Domain Indices of Heart Rate Variability (dULF)

HRV (heart rate variability), as a functionality indicator for the autonomic system, exerts an indispensable effect on the VVS pathogenesis. Wang et al. explored the utility of the HRV frequency-associated indicators dULF (daytime ultra-low-frequency), nULF (nighttime ultra-low-frequency), dVLF (daytime very-low-frequency), and nVLF (nighttime very-low-frequency) to differentially diagnose POTS and VVS. In children with VVS, the values of nVLF, dVLF, nULF, and dULF were much higher than in children with POTS, suggesting that VVS is associated with greater sympathetic excitability. Further analysis found that dULF could serve as a physiological marker to make a differential diagnosis between the two disorders as it yields a higher predictive value than the other indicators. Through the dULF value evaluation based on an ROC (receiver operating characteristic) graph, the diagnostic differentiation of VVS from POTS was achieved. Children with clinical symptoms of OI were diagnosed as having VVS if their dULF was > 36.2 ms^2^, for which the diagnostic sensitivity and specificity were 71.4% and 75.0%, respectively [21].

## 3. Individualized Therapy

It has been assumed that absolute hypovolemia, autonomic neural imbalance, peripheral vascular resistance dysregulation, and hyper-adrenergic responses are involved in OI pathogenesis [4]. Hence, children with VVS or POTS have received salt, water, beta-blockers, or midodrine as treatments. Occasionally, octreotide or pyridostigmine have been used to treat POTS patients, albeit with varying efficacies [22,23,24,25,26]. Considering their different mechanisms and the poor results of current treatments, scientists have sought improvements using individualized treatments. Great improvements were achieved in terms of individualized therapies and before the application of any treatment, biological markers or predictors could provide useful information for doctors to choose a specific treatment regimen. Hence, in the following section, we would like to summarize the recently discovered biological markers or predictors to promote optimal treatment strategies for patients with POTS and VVS. (Table 2) (Figure 1)

### 3.1. Individualized Therapy for POTS

#### 3.1.1. Physiological Indicators Predicting the Efficacy of Non-Pharmacotherapy Treatment in Children with POTS

Symptoms of POTS patients have been reported to be ameliorated by a short-period regular program of progressive physical activities [50]. These treatments include physical movement, exercise designed to enhance physical fitness, and sleep–wake loop rectification. Physical motion, such as stamping, leg crossing, and slow standing with a bowed head, help prevent the flow of blood into the lower body parts and sustain cerebral circulation during an upright posture [51,52]. Muscle weakness can be prevented and sympathetic activity increased by moderate physical training. Acting as a blood pump, leg muscles make an important contribution to improving cardiac venous return and enhancing heart output [53]. The sleep–wake pattern of children with POTS is usually disrupted, which is related to the disruption of the circadian rhythm of parasympathetic and sympathetic activation. Certain patients have very different circulatory reactions to standing in the morning and evening [54], hence, identifying patients who will respond to nonpharmacotherapy treatments requires further research.

HR-corrected QT interval dispersion (QTcd)

Autonomic dysfunction is believed to be a critical mechanism of POTS and could serve as an important therapeutic target. Treatments of POTS based on autonomic dysfunction comprise pharmacotherapy and non-pharmacotherapy treatments. Physical treatment, which is non-invasive and does not involve drug treatment, might improve symptoms, however, its effects are inconsistent [25,55]. QT interval dispersion reflects autonomic nervous function. Lu et al. observed higher values of QTcd among POTS children than among normal controls; and compared with non-responders, those who responded to treatment had a higher QTcd before treatment, and the responders had a more significant decrease in symptom scores post-treatment. The predictive significance of QTcd for therapeutic responses to the physical treatment of POTS was evaluated by exploiting an ROC graph. Analysis of the ROC graph yielded an AUC (area under the curve) of 0.73. Using a cut-off value for QTcd of 43.0 msec resulted in 90% sensitivity and 60% specificity [27]. Thus, in children with POTS, QTcd might help to predict the effects of physical treatment.

Salivary cortisol levels

Patients with POTS have a significantly decreased quality of life because of sleep problems [56]. Children who sleep for less than 8 h a day have an almost six times higher risk of developing POTS than those with a daily sleep duration of ≥8 h [57]. The underlying cause of this is poorly understood. Lin et al. reported a prominently higher salivary concentration of cortisol among POTS patients compared to the controls. In addition, children who responded to sleep-promoting therapies exhibited prominently higher salivary levels of cortisol on waking than the non-responders. In patients with POTS, the at-awakening salivary cortisol concentration could predict the effectiveness of sleep-promoting therapies. The AUC value was 75.8% with 83.3% sensitivity and 68.7% specificity [28]. The pre-awakening salivary cortisol concentration might be useful in forecasting the efficacy of sleep-promoting therapies for POTS children; a cortisol concentration >4.1 ng/mL indicates the likelihood of a clinical improvement after a sleep-promoting therapy.

#### 3.1.2. Physiological Predictors for the Oral Rehydration Salts Efficacy in Pediatric POTS

Although POTS involves heterogeneous mechanisms, its effects are similar to that of known forms of hypovolemia and involve reductions in cardiac output and systemic venous return [58,59]. In patients with POTS, the redistribution of the blood volume decreases the central blood volume; however, the total blood volume does not change, resulting in tachycardia, vagal withdrawal, and reflex sympathetic excitation [60]. Therefore, in patients with POTS, an intravenous saline or oral rehydration solution (ORS) could prevent syncope and improve OI and result in changes in the HR. However, the efficacy of salt replenishment was observed only in some POTS patients [61,62]; therefore, a worthwhile research topic would be a method for identifying responders to this therapy.

24 h urinary sodium

Hypovolemia is one of the pathogeneses of POTS. Although the underlying pathophysiology of POTS is uncertain and much effort has been made to treat cases, increasing the blood volume by elevating salt and fluid consumption has been confirmed to benefit children and adolescents with POTS. Only some patients with POTS benefit from salt supplementation; therefore, it is very important in clinical practice to identify patients who will respond to this treatment. Zhang et al. found that the urinary excretion of sodium in patients who responded to salt replenishment was low for 24 h. Under a <124 mmol/24 h threshold for the 24 h urinary excretion of sodium, the ORS efficacy forecast had a sensitivity of 76.9% and a specificity of 93% to treat POTS [29].

Changes in heart rate during head-up tilt test (HUTT).

Excessive orthostatic tachycardia is the typical hemodynamic standard for both pediatric and adolescent POTS sufferers [63]. The HUTT is a widely accepted test to evaluate POTS in children and adolescents [64,65]. Lin et al. found that heart rate changes during the HUTT differed between POTS patients who responded to ORS and those who did not respond. The ROC curve showed that the increase in the pre-therapy cardiac rate was 41 BPM (sensitivity = 72%, specificity = 70%) or the pre-therapy 10 min upright cardiac rate was 123 BPM at maximum (with 48% sensitivity 4 and 78% specificity). At this point, the HUTT is applicable to predict the response to ORS of children with POTS. When using the two indices at the same time, 84% sensitivity and 56% specificity were observed. Thus, heart rate changes during the HUTT help to forecast the ORS response among POTS children [30].

Body mass index (BMI)

Body mass index (BMI) is not only a parameter for defining degrees of overweightness and obesity, but it is also an accurate marker for identifying increased cardiovascular risk [66]. Bunsawat K et al. showed that young adults with obesity exhibited smaller postexercise peripheral vasodilation compared with young adults without obesity [67]. Since OI patients have abnormal increased peripheral vasodilation, a number of researchers have explored whether BMI is also associated with this disease, this question in depth. The BMI of patients with POTS with a low blood volume was significantly lower than that of patients with a normal blood volume. This suggested that BMI is related to blood volume [68]. ORS could increase the blood volume, thus improving symptoms. Theoretically, the efficacy of ORS can be improved if patients with hypovolemia are identified by their BMI. Li et al. explored the association between BMI and the ORS response among pediatric POTS sufferers. The results revealed that the BMI at baseline was highly sensitive and specific to forecasting the ORS response among POTS patients. The ROC curve analysis showed that the BMI cut-off value was 18.02, the sensitivity for predicting the effectiveness of ORS was 92%, and the specificity was 82.8%. Thus, ORS treatment can produce satisfactory therapeutic effects in patients with POTS with low BMI [31].

Baroreflex sensitivity (BRS)

The main pathogenesis of POTS is autonomic reflex abnormality [69,70]. Baroreflex sensitivity (BRS) is critical in the autonomic reflex process, exerting a vital function to maintain cardiovascular stability, particularly in mediating the alterations of upright blood pressure [71]. Li et al. treated each POTS child with conventional therapies after diagnosis, including ORS, autonomic function training, as well as health education, discovering prominently higher BRS in the POTS children than in the healthy controls. The area under the ROC curve for the predictive significant evaluation of BRS in the outcome of children with POTS who were treated with ORS was 0.855. A 17.01 ms/mmHg threshold of BRS resulted in 85.7% sensitivity and 87.5% specificity. Hence, the patient outcome could be predicted by measuring the BRS in patients treated by ORS. BRS determination has the advantages of being convenient, low-cost, easy to perform, and non-invasive [32].

Mean corpuscular hemoglobin concentration (MCHC)

One of the major underlying causes of POTS is hypovolemia. Changes in the RBC (red blood cell) volume and count play an essential role in POTS pathogenesis, which could be related to hypovolemia [72]. To investigate if hemocytometry indexes could qualify as predictors of ORS efficacy among pediatric POTS sufferers, Lu et al. recorded baseline hemocytometric variables and treated POTS patients with ORS for 3 months. The results showed that both larger mean corpuscular volume (MCV) and lower mean corpuscular hemoglobin concentration (MCHC) values were independent risk factors for developing POTS. In children with POTS, responders to ORS had baseline lower MCV and higher MCHC than nonresponders (*p* < 0.05). The ROC graph suggested that the AUC for predicting the value of MCHC was 0.73. An MCHC cut-off value of 347.5 g/L resulted in 68.8% sensitivity and 63.2% specificity for the prediction of ORS treatment effects on POTS [33].

#### 3.1.3. Physiological Predictors for the Midodrine Hydrochloride Efficacy among POTS Children

A crucial factor in pathophysiological POTS development is abnormal peripheral vascular constriction. Accordingly, in recent years, the application of peripheral vasoconstrictive medications has been put forward as a way of syncope relapse prevention for POTS patients. An alpha-1 adrenoreceptor agonist called midodrine hydrochloride can narrow the vascular lumen through peripheral blood pressure elevation and its efficacy in the treatment of POTS has been confirmed [73]. However, treatment using midodrine hydrochloride resulted in a response rate of only around 70% [24]. Therefore, a pre-treatment prediction of midodrine hydrochloride’s therapeutic effect on POTS would be clinically important to improve therapeutic effectiveness.

Pro-adrenomedullin (MR-proADM)

The vasoconstrictor, midodrine hydrochloride, improves symptoms and inhibits an increased HR when standing. MR-proADM and ADM are produced in equimolar amounts, which are potent vasodilators with diuretic, natriuretic, and antimitogenic effects. The plasma levels of MR-proADM were prominently higher in the pediatric POTS patients compared with the healthy controls. Among midodrine hydrochloride responders, the plasma MR-proADM level was prominently higher in contrast to the non-responders. The AUC for the forecasting effect of MR-proADM was 0.879. An MR-proADM cut-off value of 61.5 pg/mL resulted in 100% sensitivity and 71.6% specificity for predicting midodrine hydrochloride’s effectiveness in treating POTS [34].

Flow-mediated vasodilation response (FMD)

Abnormal vasodilation leads to dysregulation of peripheral vascular resistance, which exerts a pivotal effect on the POTS pathogenesis [74]. Midodrine hydrochloride constricts peripheral blood vessels and increases venous return [75]. Vascular endothelial function is assessed clinically using FMD. Children with POTS had a markedly higher FMD than the control group during the pre-treatment HUTT and those who responded to midodrine hydrochloride exhibited prominently higher pretreatment FMD compared to the non-responders (*p* < 0.05). An ROC curve assessed FMD’s predictive value for the response to midodrine hydrochloride therapy in POTS. An FMD > 9.85% resulted in a sensitivity of 76.9% and a specificity of 93% regarding predicting the efficacy of midodrine hydrochloride in treating POTS for 1 month. For effective treatment for 3 months, the sensitivity was 71.6% and the was 80% [35].

Hydrogen sulfide in erythrocyte (H_2_S)

One mechanism that leads to POTS symptoms is believed to be an excessive decrease in blood volume. Midodrine hydrochloride has vasoconstrictive activity. In patients with POTS, midodrine hydrochloride treatment reduced tachycardia in the HUTT [24]. However, midodrine hydrochloride is effective in some, but not all, patients with POTS. Therefore, it would be useful to develop a clinical method to predict a patient’s response to midodrine hydrochloride. Recently, H_2_S was revealed to be a vasodilating gasotransmitter. Children with POTS had markedly increased plasma H_2_S levels and the endogenous H_2_S was primarily released from erythrocytes [12]. Yang et al. found that the amount of H_2_S produced from erythrocytes was prominently larger in the POTS patients compared to the controls. A prominently larger H_2_S production was noted among midodrine hydrochloride responders compared to the non-responders. The AUC of the ROC curve in the evaluation of the erythrocytic level of H_2_S capable of forecasting the midodrine hydrochloride response in POTS was 0.813. An erythrocytic H_2_S level > 27.1 nmol/min/10^8^ resulted in 78.9% sensitivity and 77.8% specificity for predicting midodrine hydrochloride’s effects in children with POTS [36].

Blood pressure changes in the upright position

Mechanistically, dysfunction of vasodilation is responsible for POTS symptoms [76]. Midodrine hydrochloride has a vasoconstrictor effect, thus it was speculated that children with reduced vascular resistance and a slower response to blood pressure changes during the HUTT might respond preferably to midodrine hydrochloride. Deng et al. explored quantified blood pressure variations in the course of the HUTT to assess midodrine hydrochloride’s therapeutic efficacy among pediatric POTS sufferers. As revealed by ROC graph analysis, midodrine hydrochloride was effective when elevation in SBP (systolic blood pressure) was ≤ 0 mmHg or elevation in DBP (diastolic blood pressure) was ≤ 6.5 mmHg (supine-to-standing) prior to therapy in children with POTS. Using SBP and DBP to predict midodrine hydrochloride’s therapeutic effects resulted in 72% sensitivity and 88% specificity, with the AUC values being 0.744 and 0.809, respectively [37].

Arginine vasopressin (AVP)/Plasma copeptin

Midodrine hydrochloride could improve the clinical symptoms of POTS by inhibiting the activity of sympathetic nerves and reducing the capacity of peripheral veins [77]. The bodily secretion amount of copeptin is equimolar to that of AVP. AVP is an antidiuretic hormone that maintains hemodynamic stability, regulates osmotic pressure, and regulates the biological functions of the central nervous system (CNS) [78,79]. According to a study by Zhao et al., POTS patients exhibited prominently higher plasma concentrations of copeptin than the healthy controls, and the plasma copeptin levels in patients who responded to midodrine hydrochloride were prominently higher compared to the non-responders. Regarding the AUC for the predictive significance of plasma copeptin, its value was 0.8. A cut-off value of 10.482 pmol/L for plasma levels of copeptin resulted in 81.3% sensitivity and 76.5% specificity for predicting midodrine hydrochloride’s efficacy in treating POTS [38].

#### 3.1.4. Physiological Predictors for the Efficacy of Metoprolol in Pediatric POTS

Beta-adrenoceptor blockers are often applied in the management of pediatric POTS patients. They inhibit sympathetic nervous system activity and reduce stimulation of the heart rate and cardiac baroreceptors, thereby blocking the effects of increased blood catecholamine levels. However, only some children’s symptoms are improved using β-blockers [23,80,81]. In addition, β-blockers might affect children’s exercise tolerance [82]. Therefore, to improve therapeutic effectiveness, it is important and clinically significant to predict the efficacy of metoprolol, a β-blocker, before the treatment of POTS.

Orthostatic plasma norepinephrine

A hyperadrenergic state is one of the important mechanisms leading to POTS. Increases in orthostatic plasma norepinephrine, which might be the result of damaged baroreflex-mediated vasoconstriction, represent the key hyperadrenergic biochemical change in patients with POTS [83]. Metoprolol’s efficacy for managing pediatric POTS was associated with orthostatic plasma norepinephrine levels. Plasma norepinephrine levels in patients who responded to metoprolol were significantly higher compared with those in non-responders. The area under the ROC curve for the ability of plasma norepinephrine levels to predict metoprolol’s effects was 0.785. A norepinephrine cut-off value of 3.59 pg/mL resulted in 76.9% sensitivity and 91.7% specificity for predicting metoprolol’s treatment effect on POTS [39].

AVP/Plasma copeptin

The key pathogeneses of POTS are the hyperadrenergic status and relative hypovolemia. POTS is characterized by tachycardia and β-adrenergic inhibitors can inhibit sympathetic nervous system activity, decreasing the stimulus to the heart baroreceptor, which thus reduces the high plasma level of catecholamine, thereby relieving the tachycardia-induced discomfort at a relatively low blood volume [23,81]. Low levels of AVP might indicate hyperadrenergic conditions [76]. The secretion level of copeptin, a joint AVP glycopeptide, was probably equivalent to that of AVP and the two maintain stability during circulation [79,84]. A study reported prominently higher baseline concentrations of plasma copeptin among POTS children compared to the healthy controls, whereas the plasma copeptin levels of responders to metoprolol were lower compared to the non-responders. An AUC of 0.889 was yielded for the predictive significance of plasma copeptin. A baseline plasma copeptin cut-off value of 10.225 pmol/L resulted in 90.5% sensitivity and 78.6% specificity for predicting metoprolol’s efficacy in children with POTS. Therefore, baseline plasma copeptin levels represent a biomarker for predicting metoprolol’s effectiveness to treat pediatric POTS [40].

Plasma C-type natriuretic peptide (CNP)

CNP, which ranks third among natriuretic peptides just after ANP (atrial natriuretic peptide) and BNP (brain natriuretic peptide), elevates the mRNA expression of TH (tyrosine hydroxylase) through the cGMP/PKG axis, which then augments the synthesis of catecholamine. POTS symptoms worsen with increased plasma catecholamine levels. Therefore, Lin et al. believed that plasma CNP could reflect blood catecholamine levels and, to a certain extent, the heart rate. Their study showed that the plasma CNP was significantly higher among POTS children (51.9 ± 31.4 pg/mL) than the normal controls (25.1 ± 19.1 pg/mL) (*p* < 0.001). For the metoprolol responder patients, prominently higher pre-treatment plasma CNP was observed in contrast to the non-responders. An AUC value of 0.821 was yielded. A plasma CNP cut-off concentration of > 32.55 pg/mL resulted in a sensitivity of 95.8% and a specificity of 70% for forecasting metoprolol’s efficacy in POTS [41].

Heart rate variability (HRV)

HRV can be a crucial indicator of the vagal and sympathetic nerve functions. Wang et al. investigated if baseline HRV indicators prior to treatment with metoprolol could serve as a predictor for its anti-POTS efficacy. In responders, the HRV frequency indexes HF (high frequency) and LF (low frequency), as well as the temporal domain indices TR (triangular index) and SDNN (standard deviation for N-N intervals) were prominently lower compared with the non-responders. Long-term observation of the patients revealed that a TR index ≤ 33.7 jointly with an SDNN ≤ 79.0 ms could be valid preliminary predictors for the metoprolol response in pediatric POTS, with a sensitivity of 85.3% and a specificity of 81.8% [42].

Heart Rate (HR) and Heart Rate Difference

The HR and its discrepancies can serve as predictors for POTS patients’ responses to β-blocker treatment via a mechanism that could be associated with metoprolol-elicited repression of brainstem adrenoceptor activation and the reduction in cardiac parasympathetic activity that results in an HR increase [85,86]. Wang et al. evaluated the ability of the HR and HR difference during the HUTT, which predicted the metoprolol therapy-associated clinical improvements among pediatric and adolescent POTS sufferers. The results showed that after three months of metoprolol treatment, all patients with POTS experienced clinical improvements. The HR and its discrepancies could forecast metoprolol’s therapeutic potency for POTS in adolescents and children. An ROC curve assessment of the HR and HR difference’s predictive values for the effectiveness of metoprolol resulted in areas under the curve of 0.794 for HR 5, 0.802 for HR 10, 0.905 for HR difference 5, and 0.901 for HR difference 10. An HR 5 ≥ 110 beats/min resulted in 82.50% sensitivity and 69.23% specificity for predicting metoprolol’s effect on POTS. An HR 10 ≥ 112 beats/min resulted in 84.6% sensitivity and 69.70% specificity. An HR difference of 5 ≥ 34 beats/min resulted in 85.29% sensitivity and 89.47% specificity. An HR difference 10 ≥ 37 beats/min resulted in 97.56% sensitivity and 64.86% specificity [43].

### 3.2. VVS Individualized Therapy

#### 3.2.1. Physiological Predictors for the Orthostatic Exercise Efficacy in Pediatric VVS

The possible mechanisms of VVS include genetic factors, an abnormal Bezold–Jarisch reflex, vasomotor dysfunction, and autonomic nervous dysfunction [87,88]. Therefore, improving autonomic nervous function via orthostatic training is believed to be a treatment option in pediatric VVS. During orthostatic training, the patient stands against a wall with their feet placed 15 cm from the edge of the wall without moving. The duration of standing is increased gradually from 3 to 30 min according to the patient’s standing tolerance [27,89]. To enhance the effectiveness of orthostatic training in pediatric VVS, it is important to identify those patients with VVS in whom autonomic dysfunction might be the primary mechanism because these patients tend to respond positively to orthostatic exercise.

The Acceleration index (AI)

AI refers to an instantaneous HR fluctuation in patients that happens during a supine-to-upright transition of posture reflecting sympathetic nerve activity. Sundkvist et al. observed a positive correlation between the AI and plasma concentration of epinephrine during the initial 60 s following standing, which prompted the suggestion that the AI might represent sympathetic activity [90]. Tao et al. found that the mean AI was significantly lower in responders before treatment compared with non-responders. Using an ROC graph, the AI’s predictive significance for the response to orthostatic training therapy in VVS was evaluated. An AUC of 0.827 was yielded and an AI cut-off value of 26.77 resulted in 85.0% sensitivity and 69.2% specificity [44].

#### 3.2.2. Physiological Indicators to Predict the Efficacy of a1-Adrenergic Receptor Agonists in Pediatric VVS

The selective alpha-adrenergic receptor agonist midodrine has peripheral action that increases peripheral vascular resistance and reduces venous pooling [91]. Vasodilation has a primary role in the pathogenesis of VVS; therefore, peripheral alpha-agonists were believed to decrease susceptibility to VVS. In addition, midodrine does not penetrate the blood–brain barrier, resulting in no CNS side effects. Midodrine is an approved agent for orthostatic hypotension management in the United States [92]. The beneficial effects of midodrine have been confirmed in a series of studies on adult VVS; however, most of these studies were not randomized, double-blind, or placebo-controlled studies [93,94].

Flow-mediated vasodilation (FMD)

Vascular endothelial dysfunction is one of the mechanisms of VVS [95]. Brachial arterial FMD is capable of evaluating vascular endothelial function using Doppler ultrasound. Midodrine hydrochloride might play a therapeutic role by affecting the functionality of vascular endothelium among pediatric VVS patients with enhanced FMD. Zhang et al. explored whether the brachial arterial FMD could act as a predictor for midodrine hydrochloride’s therapeutic efficacy in pediatric VVS. They found significantly higher FMD levels in patients who experienced good therapeutic efficacy than those with poor therapeutic efficacy. The AUC for the predictive significance of FMD was 0.895. An FMD cut-off of 8.85% resulted in high sensitivity (90%) and specificity (80%) for predicting midodrine hydrochloride’s efficacy in VVS treatments [45].

#### 3.2.3. Physiological Indicators to Predict β-Blocker Efficacy in Pediatric VVS

β-blockers have been used for many years as a treatment for VVS. The proposed mechanism involves a decrease in left ventricular mechanoreceptor activation, which is regarded as a sympathetic tone reduction and serum adrenaline elevation before syncope [96]. Several uncontrolled studies reported that β-blockers were effective; however, in five of seven controlled studies, they were not beneficial [97,98,99,100,101,102,103]. Chen, et al. demonstrated that 60.61% of children with VVS treated using oral metoprolol were cured, whereas a Russian study reported an overall efficacy of only 44% after atenolol treatment [104,105]. These results suggested the probable validity of β-blockers for some particular VVS patients and it is necessary to predict the response to β-blockers so that their reasonable application can be justified.

Heart rate (HR)

To determine the predictors for the efficacy of metoprolol in pediatric VVS, the hemodynamic traits were examined by Zhang et al. in the course of a HUTT. The results showed that the increase in the HR during the HUTT among the metoprolol responder patients was prominently greater compared to those who did not respond ((42 ± 16) beats/min vs. (18 ± 13) beats/min, *p* < 0.01). A 30-BPM elevation in the HR above the baseline combined with a positive HUTT response resulted in 81% sensitivity and 80% specificity regarding the validity of forecasting metoprolol’s efficacy in VVS [46].

LVFS (left ventricular fractional shortening) and LVEF (left ventricular ejection fraction)

Previous studies have shown the correlations of VVS caused by tilt testing with elevations in the adrenaline, noradrenaline, cAMP, and dopamine levels. This indicated that adrenal sympathetic stimulation occurs before the vasovagal syncopal development, which is probably reflected in its pathophysiology [106]. Certain patients with VVS experience bradycardia and reflex hypotension, which might be induced by excessive ventricular constriction associated with increased catecholamines, providing the rationale for the use of β-blockers. Studies have shown that LVEF and LVFS might be related to plasma catecholamine levels [107]. Song et al. explored whether these two measures were capable of forecasting metoprolol’s efficacy in pediatric VVS. A six-month follow-up assessment showed that those who responded to treatment with metoprolol had higher baseline LVFS and LVEF values than those who did not (LVFS: 41.1 ± 1.9% vs. 35.8 ± 3.6%, *p* = 0.002; LVEF: 72.8 ± 2.8% vs. 65.5 ± 4.6%, *p* = 0.001). The ability of LVFS and LVEF to forecast VVS patients’ responses to metoprolol treatment was assessed using an ROC curve. At the two-month treatment point when the LVEF was > 70.5%, sensitivity was 80.0% and specificity was 100.0%; when the LVFS was > 38.5%, sensitivity was 90.0%, and specificity was 90.0%. At the six-month treatment point when the LVEF was > 70.5%, sensitivity was 81.3%, and specificity was 88.9%; when the LVFS was > 37.5%, sensitivity was 93.8%, and specificity was 66.7% [47]. However, an LVEF cut-off value of 70.5% might exclude some patients from the main populations, which limits the utility of LVEF for forecasting the VVS patients’ responses to the metoprolol therapy.

Baroreflex sensitivity (BRS)

The baroreceptor reflex (BR) counts as the foremost nervous regulatory reflex for keeping homeostasis between blood flow and pressure and has a vital function in the changes in blood flow between different postures. Patients with VVS have baroreceptor dysfunction and are prone to syncope when external stimulation leads to abnormal regulation of the nervous system [108]. β-blockers reduce the recurrence of syncope by reducing the stimulation of the baroreceptors [75]. However, the effects of treatment vary [26,109,110]. Measuring the baroreflex sensitivity (BRS) enables the evaluation of baroreceptor responses to alterations in blood pressure and reflects the influence of vagus nerve activity on the heart. Thus, the response of patients with VVS to treatment using beta-blockers could be predicted using the BRS as a biomarker. Tao et al. found that compared with non-responders, patients who responded to treatment had a markedly elevated supine BRS value. The responders (8.0 ± 7.8 ms/mmHg) exhibited more evident BRS alterations compared to the non-responders (−3.0 ± 10.4 ms/mmHg) (*p* < 0.01). An ROC curve evaluation of supine BRS to predict the response to metoprolol treatment in patients with VVS identified a threshold of 10 ms/mmHg, with 82% sensitivity and 83% specificity. Using a 4 ms/mmHg threshold led to a sensitivity decline to 71%, however, the specificity remained unchanged [48].

24 h urine norepinephrine (24 h urine NE)

Abnormal sympathetic activity exerts an indispensable impact on the VVS pathogenesis [111]. Syncope might occur in upright positions when sympathetic compensatory activity is reduced [112]. β-blockers are effective at hindering the activity of the circulatory high concentrations of catecholamines; however, their effectiveness was inconsistent [103,113,114], which suggests that children with VVS have different baseline sympathetic activities. A study observed that metoprolol responders (40.75 ± 12.86 μg/24 h) had prominently higher 24 h urinary NE concentrations compared to non-responders (21.48 ± 6.49 μg/24 h) (*p* < 0.001). An ROC curve was used to assess the ability of 24 h urine NE to forecast VVS sufferers’ responses to the metoprolol therapy. With a 34.84 μg/24 h threshold for the 24 h urinary NE, both the sensitivity (70%) and specificity (100%) were high for predicting the effectiveness of metoprolol in the treatment of VVS [49].

## 4. Conclusions

The strategies and methods used to treat VVS and POTS are different and it is important to clearly distinguish the two diseases. Currently, POTS and VVS are diagnosed based mainly on clinical symptoms and positive changes in the HUTT. However, the hemodynamic changes during the HUTT sometimes overlap, thus obscuring the diagnosis. Ongoing research has identified methods to differentiate VVS and POTS, which are useful in clinical practice. HRV, AI, the 30/15 ratio, plasma H_2_S, and the serum iron level are all considered useful indexes. However, we believe that in the future, better indicators will be developed to differentiate between these two clinical syndromes.

Significant progress has been made in the individualized management of pediatric POTS and VVS. Hemodynamic parameters or biomarkers can be used as objective indexes for POTS and VVS in pediatric populations and they are capable of guiding the appropriate therapy selection. Unfortunately, the predictive value of the markers or indexes varies and none of them can be regarded as the gold standard. Indeed, more sensitive predictive indicators for the effectiveness of different treatment modalities and better methods of diagnosis are required to improve the individualized management of POTS and VVS in children and adolescents.

## Figures and Tables

**Figure 1 ijerph-19-06974-f001:**
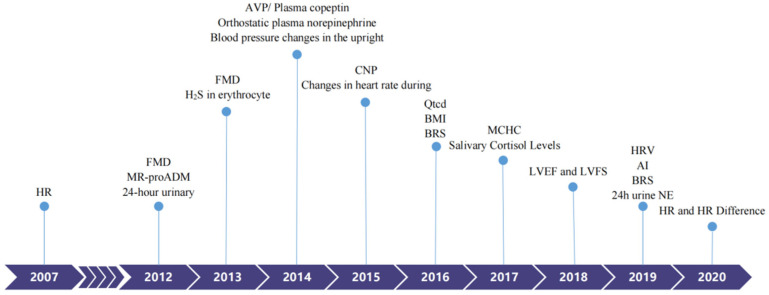
Biomarkers to predict individualized treatment for VVS and POTS in chronological order. POTS: Postural Tachycardia Syndrome; VVS: Vasovagal Syncope; HR: heart rate; FMD: Flow-mediated vasodilation response; MR-proADM: Pro-adrenomedullin; AVP: Arginine vasopressin; CNP: C-type natriuretic peptide; BMI: Body mass index; BRS: Baroreflex sensitivity; MCHC: Mean corpuscular hemoglobin concentration; LVEF: Left ventricular ejection fraction; LVFS: Left ventricular fractional shortening; AI: Acceleration index; HRV: Heart rate variability; 24 h urine NE: 24-h urine norepinephrine.

**Table 1 ijerph-19-06974-t001:** Clinical indicators used to differentiate POTS from VVS.

	Cut-Off	Sensitivity	Specificity	Year
Plasma H_2_S level	98 μmol/L	90%	80%	2012
Serum iron level	11.8 μmol/L,	92.50%	64.70%	2013
AI and 30/15	AI: 28.180; 30/15: 1.025	95.80%	80.80%	2018
dULF	36.2 ms^2^	71.40%	75.00%	2019

POTS: postural tachycardia syndrome; VVS: vasovagal syncope; AI: Acceleration Index; dULF: daytime ultra-low frequency; 30/15: ratio of the R-R interval length between the 30th and 15th beats in the upright position.

**Table 2 ijerph-19-06974-t002:** Clinical indicators used to predict individualized treatment of POTS and VVS.

Diagnosis	Treatment	Biological Markers or Predictors	Cut-off		Sensitivity	Specificity	Year
POTS	non-pharmacotherapy	Qtcd [27]	43.0 msec		90%	60%	2016
		Salivary cortisol levels [28]	4.1 ng/mL		83.30%	68.70%	2017
	ORS	24-h urinary sodium [29]	124 mmol/24 h		76.90%	93%	2012
		Changes in heart rate during HUTT [30]	pre-treatment increase in HR = 41 beats/min maximum upright HR in 10 min = 123 beats/min		84%	56%	2015
		BMI [31]	18.02		92%	82.80%	2016
		BRS [32]	17.01 ms/mmHg		85.70%	87.50%	2016
		MCHC [33]	347.5 g/L		68.80%	63.20%	2017
	midodrine hydrochloride	MR-proADM [34]	61.5 pg/mL		100%	71.60%	2012
		FMD [35]	9.85%	1-month	76.9%	93%	2013
				3-month	71.6%	80%	
		H2S in erythrocyte [36]	27.1 nmol/min/10^8^		78.90%	77.80%	2013
		Blood pressure changes in the upright position [37]	SBP ≤ 0 mmHg; DBP ≤ 6.5 mmHg		72%	88%	2014
		AVP/Plasma copeptin [38]	10.482 pmol/L		81.30%	76.50%	2014
	metoprolol	Orthostatic plasma norepinephrine [39]	3.59 pg/mL		76.90%	91.70%	2014
		AVP/Plasma copeptin [40]	10.225 pmol/L		90.50%	78.60%	2014
		CNP [41]	32.55 pg/mL		95.80%	70%	2015
		HRV [42]	TR index ≤ 33.7; SDNN index ≤ 79.0ms		85.3%,	81.80%	2019
		HR and HR Difference [43]	HR 5 ≥ 110 beats/min		82.50%	69.23%	2020
			HR 10 ≥ 112 beats/min		84.62%	69.70%	
			HR difference 5 ≥ 34 beats/min		85.29%	89.47%	
			HR difference 10 ≥ 37 beats/min		97.56%	64.86%	
VVS	orthostatic training	AI [44]	26.77		85.00%	69.20%	2019
	midodrine hydrochloride	FMD [45]	8.85%		90%	80%	2012
	metoprolol	HR [46]	increase of 30 beats/min		81%	80%	2007
		LVEF and LVFS [47]	two month	LVEF > 70.5%	80.00%	100.00%	2018
				LVFS > 38.5%	90.00%	90.00%	
			six month	LVEF > 70.5%	81.30%	88.90%	
				LVFS > 37.5%	93.80%	66.70%	
		BRS [48]	10 ms/mmHg		82%	83%	2019
		24 h urine NE [49]	34.84 μg/24h		70%	100%	2019

POTS: Postural Tachycardia Syndrome; VVS: Vasovagal Syncope; Qtcd: HR-corrected QTd; ORS: Oral rehydration salts; HUTT: Head-up test; BMI: Body mass index; BRS: Baroreflex sensitivity; MCHC: Mean corpuscular hemoglobin concentration; MR-proADM: Pro-adrenomedullin; FMD: Flow-mediated vasodilation response; H_2_S: Hydrogen sulfide; SBP: Systolic blood pressure; DBP: Diastolic blood pressure; AVP: Arginine vasopressin; CNP: C-type natriuretic peptide; HRV: Heart rate variability; TR index: Triangular index; SDNN index: Standard deviation of all NN intervals; HR: Heart rate; AI: Acceleration index; LVEF: Left ventricular ejection fraction; LVFS: Left ventricular fractional shortening; 24 h urine NE: 24 h urine norepinephrine.

## Data Availability

Not applicable.

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
