# Peer review of "Biomarkers and Hemodynamic Parameters in the Diagnosis and Treatment of Children with Postural Tachycardia Syndrome and Vasovagal Syncope"

_ijerph, 2022, doi:10.3390/ijerph19126974_

Round 1

Reviewer 1 Report

In this paper, authors aimed to summarize the current state of research into biomarkers for distinguishing the diagnosis of pediatric vasovagal 
syncope from that of postural tachycardia syndrome. It is already known that orthostatic intolerance is very common in young patients and that absolute hypovolemia, autonomic neural imbalance, peripheral vascular resistance dysregulation, and hyper-adrenergic responses are involved.

I would like to suggest only to spend more words on body mass index and its influence on orthostatic intolerance, as well expressed here: "Antonini-Canterin F.  Obesity, Cardiac Remodeling, and Metabolic Profile: Validation of a New Simple Index beyond Body Mass Index. J Cardiovasc Echogr. 2018 Jan-Mar;28(1):18-25. doi: 10.4103/jcecho.jcecho_63_17" and here: "Bunsawat K, Central and Peripheral Postexercise Blood Pressure and Vascular Responses in Young Adults with Obesity. Med Sci Sports Exerc. 2021 May 1;53(5):994-1002. doi: 10.1249/MSS.0000000000002540"

Author Response

Dear Reviewer,

We appreciate the comments raised by you. We have carefully considered these comments and addressed them in the revised manuscript. A point-by-point response is provided below. Please check the attached document which contains the opinions of all the reviewers.

Reviewer:  In this paper, authors aimed to summarize the current state of research into biomarkers for distinguishing the diagnosis of pediatric vasovagal syncope from that of postural tachycardia syndrome. It is already known that orthostatic intolerance is very common in young patients and that absolute hypovolemia, autonomic neural imbalance, peripheral vascular resistance dysregulation, and hyper-adrenergic responses are involved.

I would like to suggest only to spend more words on body mass index and its influence on orthostatic intolerance, as well expressed here: "Antonini-Canterin F.  Obesity, Cardiac Remodeling, and Metabolic Profile: Validation of a New Simple Index beyond Body Mass Index. J Cardiovasc Echogr. 2018 Jan-Mar;28(1):18-25. doi: 10.4103/jcecho.jcecho_63_17" and here: "Bunsawat K, Central and Peripheral Postexercise Blood Pressure and Vascular Responses in Young Adults with Obesity. Med Sci Sports Exerc. 2021 May 1;53(5):994-1002. doi: 10.1249/MSS.0000000000002540"

Response:Thank you for your suggestion. We have add more information to the BMI section in the revised version as you suggested. At the same time, the two references you mentioned have also been added to the references list (ref 47 and ref 48).

Please see Page 7, Line 245: Body mass index (BMI) is not only a parameter to define the degrees of overweight and obesity, but also an accurate marker in identifying increased cardiovascular risk[47]. Bunsawat K et al showed that young adults with obesity exhibited smaller postexercise peripheral vasodilation compared with young adults without obesity[48]. Since the OI patients have an abnormal increased peripheral vasodilation, so whether the BMI were also associated with this disease, a number of research have explored this question in depth.

Please see Page 16 , Line 705 :  

  1. Antonini-Canterin, F.; Di Nora, C.; Poli, S.; Sparacino, L.; Cosei, I.; Ravasel, A.; et al. Obesity, Cardiac Remodeling, and Metabolic Profile: Validation of a New Simple Index beyond Body Mass Index. J Cardiovasc Echogr 2018, 28, 18–25. doi:10.4103/jcecho.jcecho_63_17
  2. Bunsawat, K.; Lefferts, E. C.; Grigoriadis, G.; Wee, S. O.; Kilianek, M. M.; Fadel, P. J.; et al. Central and Peripheral Postexercise Blood Pressure and Vascular Responses in Young Adults with Obesity. Med Sci Sports Exerc 2021, 53, 994–1002. doi:10.1249/MSS.0000000000002540

Reviewer 2 Report

This is a comprehensive review on the currently very actual topic the vasovagal syncope and postural tachycardia. As the authors mention, the diagnosis and differential diagnosis of VVS and POTS are much more difficult in children as compared to adults. 

Some minor comments: 

There are several repetitions of the statements and citations; eg. section 3.1.3. midodrine response rate around 70% (ref 24), section: pro-ADM: midodrine response rate around 68.4% (Ref 24).

Or statements with different goals of the review:

Section 2: In this review, we summarize the physiological indicators used to differentially diagnose POTS and VVS

Section 1: this article reviews the latest advances in diagnostically distinguishing VVS from POTS,

Table 2: References should be included

The Section 3.1. and 3.2. contains several overlapping information, while Table 2 summarizes the biological markers and predictors. Accordingly, section 3.1. and 3.2 should be reconstructed (re-written) to avoid the redundancies. 

Author Response

Dear Reviewer,

We appreciate the comments raised by you. We have carefully considered these comments and addressed them in the revised manuscript. A point-by-point response is provided below. Please check the attached document which contains the opinions of all the reviewers.

There are several repetitions of the statements and citations; eg. section 3.1.3. midodrine response rate around 70% (ref 24), section: pro-ADM: midodrine response rate around 68.4% (Ref 24). 

Or statements with different goals of the review:

Section 2: In this review, we summarize the physiological indicators used to differentially diagnose POTS and VVS

Section 1: this article reviews the latest advances in diagnostically distinguishing VVS from POTS,

Response:We appreciate your positive comments and thank you for your suggestion. It is true that we have repeated descriptions as you said. We have removed the duplicate description, and some inconsistent descriptions have been corrected.

Please see Page 8, Line 296: “However, treatment used midodrine hydrochloride resulted in a response rate of only around 70%”

         Page 8, Line 301: “The vasoconstrictor, midodrine hydrochloride, improves symptoms and inhibits the increased HR when standing. MR-proADM and ADM are produced in equimolar amounts, which are potent vasodilators with diuretic, natriuretic, and antimitogenic effects. ”

Please see Page 2, Line 48: “Hence, this article summarize the physiological indicators used to deferentially diagnose POTS and VVS, and their individualized treatment in children and discusses further research directions in this field.”

         Page 2, Line 61: “In this review, firstly, we summarize the physiological indicators used to differentially diagnose POTS and VVS (Table 1). ”

         Page 4, Line 142: “Hence, in the following section, we would like to summarize the recently discovered biological markers or predictors to promote optimal treatment strategies for patients with POTS and VVS.(Table 2) (Fig. 1)”

Table 2: References should be included

Response:Thanks for picking up this point. We couldn't agree with you more. We have added  the corresponding references in Table 2.

Please see Page 4, Table 2: We added the references the the third column. We show references right after each marker.

The Section 3.1. and 3.2. contains several overlapping information, while Table 2 summarizes the biological markers and predictors. Accordingly, section 3.1. and 3.2 should be reconstructed (re-written) to avoid the redundancies. 

Response:Thank you for your suggestion. Indeed, as you said, some of the parameters in the two parts are repeated. Some of the indicators predicted both POTS and VVS, but they predicted different treatments. For example, the BRS could be used to predicted the ORS treatment in POTS patients but midodrine hydrochloride treatment in VVS patients. Some indicators predicted different treatment in the same patients. For example, the changes of heart rate in HUTT used to predicted the ORS treatment and metoprolol in POTS patients, while AVP/Plasma copeptin used to predicted the midodrine hydrochloride treatment and metoprolol in POTS patients.

Since the section 3.1 and 3.2 are Individualized therapy for POTS and Individualized therapy for VVS respectively. In each section, we present the biomarkers or hemodynamic parameter of individualized therapy in a certain order: the index used for non-pharmacotherapy treatment (3.1.1 and 3.2.1), used for the oral rehydration salts treatment (3.1.2), used for midodrine hydrochloride treatment (3.1.3 and 3.2.2) and used for metoprolol treatment(3.1.4 and 3.2.3). After thinking carefully, we hope to keep the current order and format. On the one hand, the physiological function of each indicator in a certain treatment is described separately, on the other hand, the Table 2 is consistent with current description which is much easier for those reader who have no time to read the whole article but could understand the main information of this manuscript quickly and clearly.

We sincerely hope that you will consider our opinion.

Reviewer 3 Report

Dear Authors,

You made a good comprehensive review on Dx & Tx of POTS & VVS. I think your manuscript is good, but it needs minor revision. I wrote some recommendations on your manuscript, so please check the attached file.  Yellow-colored parts are to be revised, so please check attached memo boxes. Red-lined parts are to be deleted.

Thank you.

Author Response

Dear Reviewer,

We appreciate the comments raised by you. We have carefully considered these comments and addressed them in the revised manuscript. A point-by-point response is provided below. Please check the attached document which contains the opinions of all the reviewers.

Reviewer: You made a good comprehensive review on Dx & Tx of POTS & VVS. I think your manuscript is good, but it needs minor revision. I wrote some recommendations on your manuscript, so please check the attached file. Yellow-colored parts are to be revised, so please check attached memo boxes. Red-lined parts are to be deleted.

Response: We appreciate your positive comments and thank you so much for your correction of the mistakes in the article. We have corrected it in the revised version as suggested. The red-lined parts were all deleted while the yellow-colored parts were revised. As for the ref No. 92 which was ref No. 90 in the previously version, it is really our mistake that were wrongly cited. We replaced the incorrect citation with a new reference: Chen LY, Shen WK. Neurocardiogenic syncope: latest pharmacological therapies. Expert Opin Pharmacother 2006 Jun;7(9):1151-62. doi: 10.1517/14656566.7.9.1151.

Please see Page 2, Line 64: Table 1. Clinical indicators used to differentiate POTS from VVS

Please see Page 2, Line 71: H2S contributes to endothelium-dependent vasorelaxation and exerts regulatory effects on the pathogenesis of various diseases.

Please see Page 2, Line 75:The results of that study indicated that children with POTS and VVS had higher H2S plasma levels than healthy children. Plasma level of H2S at 98 μmol/L taken as the cutoff value produced both high sensitivity (90%) and specificity (80%) rates of correctly discriminating between patients with VVS and patients with POTS.

Please see Page 3, Line 137:Great improvement was archived in terms of an individualized therapy and before the application of any treatment, biological markers or predictors could provide useful information for doctors to choose the specific treatment regimen.

Please see Page 4, Line 145: Table 2. Clinical indicators used to predict individualized treatment of POTS and VVS

Please see Page 7, Line 228: ...low for 24 h.

Please see Page 7, Line 234: Lin et al. found that heart rate changes during the HUTT differed between POTS patients who responded to ORS and those who did not respond.

Please see Page 7, Line 240: At this point, HUTT is...

Please see Page 7, Line 270: ...who were treated  

Please see Page 7, Line 272: Hence, patient outcome could be predicted by measuring the BRS in patients treated by ORS. BRS determination has advantages of convenience, low cost, easy performance, and non-invasiveness.

Please see Page 8, Line 298: ... would be clinically important to improve ...

Please see Page 8, Line 305: ...were prominently higher in ...

Please see Page 8, Line 308: The AUC for the forecasting effect of MR-proADM was 0.879. 

Please see Page 8, Line 321:... regarding prediction of the efficacy of midodrine ... 

Please see Page 8, Line 330:... a clinical method to predict patient's ... 

Please see Page 8, Line 334: ... in the POTS patients compared to...

Please see Page 8, Line 340: ... effect in children with POTS...

Please see Page 9, Line 348: ...midodrine hydrochloride was effective when elevation in SBP (systolic blood pressure) was £ 0 mmHg or elevation in DBP (diastolic blood pressure) was£ 6.5 mmHg (supine-to-standing) prior to therapy in children with POTS.

Please see Page 9, Line 367: Physiological predictors for the efficacy of metoprolol in pediatric POTS

Please see Page 9, Line 370:They inhibit sympathetic nervous system activity and reduce stimulation of the heart rate and cardiac baroreceptors, thereby blocking the effect of increased blood catecholamine levels.

Please see Page 9, Line 374: Therefore, to improve therapeutic effectiveness, it is important and clinically significant to predict the efficacy of metoprolol, a β-blocker, before the treatment of POTS.

Please see Page 9, Line 386: ... predict metoprolol’s effects was 0.785.

Please see Page 9, Line 391: POTS is characterized by tachycardia, and β-adrenergic inhibitor...

Please see Page 10, Line 402: A baseline plasma copeptin cut-off value of...

Please see Page 10, Line 407: CNP, which ranks the third...  

Please see Page 10, Line 412:...and the heart rate somewhat.Their results study showed...

Please see Page 10, Line 416:A plasma CNP cut-off concentration of > 32.55 pg/ml resulted in a sensitivity of 95.8% and a specificity of 70% for forecasting metoprolol’s efficacy in POTS.

Please see Page 10, Line 418: ...for forecasting metoprolol’s efficacy in POTS.

Please see Page 10, Line 421:HRV can be a crucial indicator of the vagal and sympathetic nerve functions.

Please see Page 11, Line 436:...therapy-associated clinical improvement amelioration among pediatric and adolescent POTS sufferers. The results showed that after three months of metoprolol treatment, all patients with POTS experienced clinical disease improvement. The HR and its discrepancy could forecast facilitate the forecasting of metoprolol’s therapeutic potency for POTS in adolescents and children. ROC curve assessment of HR and HR difference’s predictive value for the effectiveness of metoprolol resulted in the areas under the curve of 0.794 for HR 5, 0.802 for HR 10, 0.905 for HR difference 5, and 0.901 for HR difference 10. An HR 5 ³110 beats/min resulted in 82.50% sensitivity and...

Please see Page 11, Line 454: ...from the edge of the wall...

Please see Page 11, Line 462:...supin-to-upright transition of posture,...

Please see Page 11, Line 465:Tao et al. found that the mean AI was significantly lower in responders before treatment, compared with non-responders.

Please see Page 11, Line 474:The selective alpha-adrenergic receptor...

Please see Page 12, Line 507:To determine predictors for the efficacy of metoprolol in pediatric VVS, the hemodynamic traits were examined by Zhang et al. in the course of HUTT.

Please see Page 12, Line 512:A 30-BPM elevation of HR above the baseline combined with a positive HUTT response resulted in 81% sensitivity and 80% specificity regarding the validity forecasting metoprolol’s efficacy in VVS. 

Please see Page 12, Line 524:  Studies have shown that LVEF and ...

Please see Page 12, Line 526: ...were capable of forecasting metoprolol’s efficacy...

Please see Page 12, Line 537: ...might exclude some patients from...

Please see Page 13, Line 541:...blood flow and pressure...

Please see Page 13, Line 544: ...regulation of the nervous system...

Please see Page 13, Line 550: ...compared with non-responders, ...

Please see Page 13, Line 555: ...with 82% sensitivity...

Please see Page 13, Line 562:...circulatory high concentrations of catecholamines, however,  their effectiveness was inconsistent.

Please see Page 13, Line 566:...compared to those who failed to respond (21.48 ± 6.49μg/24 h) (P < 0.001). A ROC curve was used to assess the ability of 24-h urine NE to forecast the VVS sufferers’ response to the metoprolol therapy. With a 34.84μg/24 h threshold of the 24h urinary NE,...   

Please see Page 13, Line 583: Hemodynamic parameters or biomarkers can be used as objective indexes for POTS and VVS in the pediatric population, which are capable of guiding the appropriate therapy selection. Unfortunately, the predictive value of the markers or indexes vary and none of them can be regarded as the gold standard. Indeed, more sensitive predictive indicators for the effectiveness of different treatment modalities and better methods of diagnosis are required to improve the individualized management of POTS and VVS in children and adolescents.

Please see Page 17, Line 809:  ref 92: Chen, L.Y.; Shen, W.K. Neurocardiogenic syncope: latest pharmacological therapies. Expert Opin Pharmacother 2006,7,1151-62. doi: 10.1517/14656566.7.9.1151.

Round 2

Reviewer 1 Report

The revised paper has been much improved after the last revisions.